# Adherence to the Mediterranean Food Pattern and Carbon Footprint of Food Intake by Employees of a University Setting in Portugal

**DOI:** 10.3390/nu16050635

**Published:** 2024-02-24

**Authors:** Lúcia Figueiredo, João P. M. Lima, Ada Rocha

**Affiliations:** 1Faculty of Nutrition and Food Sciences, University of Porto, 4099-002 Porto, Portugal; luciamlfigueiredo@gmail.com (L.F.); adarocha@fcna.up.pt (A.R.); 2H&TRC—Health & Technology Research Center, Coimbra Health School, Polytechnic University of Coimbra, 3045-093 Coimbra, Portugal; 3GreenUPorto—Sustainable Agrifood Production Research Centre/Inov4Agro, 4169-007 Porto, Portugal

**Keywords:** Mediterranean diet, food consumption, carbon footprint

## Abstract

Background: The Mediterranean diet is characterized by the predominance of the consumption of fruit, vegetables, cereals, nuts, and pulses; consumption of olive oil as the main source of fat; regular but moderate consumption of win; moderate consumption of fish, dairy products, eggs, and white meats; and low consumption of red meat as well as sugar and sugary products. In addition to the reported health benefits, the Mediterranean diet has also been widely recognized as a sustainable food pattern. The objective of this study was to understand the relationship between the degree of adherence to the Mediterranean diet of employees of the University of Porto and the relationship with the carbon footprint of their food consumption. Methods: An observational, analytical study was carried out, through the analysis of food consumption data collected in the form of a survey in the previous 24 h inserted in the eAT24 software, as well as the analysis of sociodemographic, lifestyle, and health data collected in the questionnaire. The carbon footprint was calculated from the previous 24 h surveys using data available on a website, obtained in carbon dioxide equivalent per kilogram of food. Sociodemographic, eating habit, and lifestyle questions were selected to understand the relationship between these and adherence to the Mediterranean diet and the carbon footprint of food consumption. Results: In total, 13.2% of the participants had a Mediterranean diet score equal to or greater than six, and the sample had an average food intake carbon footprint of 8146 ± 3081 CO_2_eq/Kg of food. A weak, statistically significant positive correlation (r = 0.142, *p* = 0.006) was observed between Mediterranean diet adherence and the carbon footprint of food intake. Conclusions: Most of the participants had a low adherence to the Mediterranean diet, as well as a high carbon footprint of food intake when compared to other countries. People with higher scores for Mediterranean diet adherence have, on average, a higher carbon footprint from food consumption intake.

## 1. Introduction

Food production has increased along with world population growth, which, in turn, has increasingly pressured the environment and its resources, mainly affecting low-income countries [1]. The extreme effects of climate change have been increasingly felt and have demonstrated the great vulnerability of ecosystems and food systems [2]. Loss of biodiversity and the degradation of soils and water quality are having increasingly serious, and sometimes irreversible, effects on food systems [3]. On the other hand, food production is the main cause of greenhouse gas emissions into the atmosphere, with agriculture accounting for 10–12% of total greenhouse gas emissions. Reducing greenhouse gas emissions is essential and urgent to ensure access to food for the next generations and requires changes in the behavior of populations, reinforced by measures taken by governments, international cooperation, and industries [4]. Healthy dietary habits based on sustainable food systems can make a difference in reducing food waste, avoiding losses along the food chain, and improving food production [5].

The Mediterranean diet (MD) presents a lower ecological footprint compared to the Western dietary pattern [6,7,8] due to the predominance of plant-based foods, local and seasonal crops, and a sustainable food culture that embraces important physical, sociocultural, economic, and environmental benefits [9]; this is the reason why the adoption of the MD triggers local food production. In 2010, the Food and Agriculture Organization of the United Nations (FAO) considered the MD as a sustainable model [10] and, in the same year, the Organization of the United Nations for Education, Science and Culture (UNESCO) considered the MD as Intangible Heritage of Humanity [11]. This distinction defined MD from the perspective of various cultural, social, and human dimensions, involving “a set of skills, knowledge, rituals, symbols, and traditions related to harvesting, fishing, animal production, conservation, processing, cooking and particularly sharing and food consumption” [12]. MD also considers cultural and lifestyle behaviors, such as moderation of servings, socialization as a basis for the feeling of belonging to a community, and conviviality associated with meal times, regular practice of physical activity, good sleeping habits, seasonality, and the consumption of minimally processed products, as well as the consumption of local and traditional products [12].

In Portugal, the MD food guide has the shape of a wheel, adapted to the pre-existing format in this country, and has introduced changes like increasing time dedicated to leisure activities, encouraging the preparation of food as a time to share with family or friends, the promotion of culinary knowledge between generations, respect for local and seasonal foods, and replacement of salt by aromatic plants when cooking [9]. In 2020, a study concluded that the Portuguese population had increased this MD adherence by 15% since 2017, but only 26% had high adherence to this dietary pattern, highlighting that the consumption of pulses, vegetables, fruits, and nuts is the most difficult recommendation to achieve [13]. The same study also reflected that the Portuguese population seems to know the MD, but both knowledge and adherence to this dietary pattern seem to be restricted to populations with higher education and income [13]. Besides its health benefits, the MD has also been recognized as a sustainable food pattern [7,14,15] that was already represented in the inverted pyramid, showing that foods recommended for more frequent consumption are associated with a lower environmental impact compared to those that should be consumed less frequently [16]. This presentation was a result of a study that concluded that infographic tools are understandable by the population, along with food education activities and healthy food choices available in food services, might simultaneously promote healthy eating habits and a lower environmental impact [16].

The evaluation of MD adherence allows the study of the food habits of the population, consequently promoting a food environment that favors a healthy lifestyle. The active adult population depends a lot on the food supply in their workplace to consolidate their eating habits and, therefore, adopt a healthy lifestyle. Lack of time can also represent a barrier to adopting a healthy diet [17], which is why food services represent an important strategy to promote health. The working population of the University of Porto (UP), in addition to having a stable economic situation, is expected to have rooted eating habits and to play an active role in promoting healthy habits, both as a working population in general and as members of the academic community of the institution. On the other hand, the study of the food intake CF can be compared with other results obtained in the literature, regarding its relationship with the MD, and it can identify practices and design interventions with the aim of reducing this environmental impact. It is also pertinent to point out that food made available in food service units and catering services that promotes the MD principles can constitute a fundamental strategy to minimize the negative impacts on the environment that these places, by themselves, already represent. Additionally, the food available in the workplace and academic environments may also represent an important strategy for promoting healthy eating habits, in particular food based on the MD recommendations.

In a representative sample of university employees, the aims of this study were as follows:
To evaluate the Mediterranean diet adherence of food consumption;To analyze the carbon footprint of food consumption;To understand the relationship between MD adherence and the CF of their food intake in this population.

## 2. Materials and Methods

An observational, analytical study was carried out [18]. The population under study is the human resources of the UP which, according to the time of data collection, accounted for 3307.7 full-time workers, of whom 52.9% were professors/researchers, 46.9% were non-teachers/non-researchers, and 0.2% were members of government bodies (Rector and Vice-Rectors) [19]. To have a representative sample of this population, it was considered necessary to have 510 individuals in this sample (13.8% of the population) to be able to find differences between averages of 25% of the standard deviation. This study included 533 workers distributed across the 14 Faculties and three Autonomous Services, constituting a representative sample of the study population. Of the total number of participants, 368 workers were considered eligible for this study because they reported a normal day of consumption in the survey 24 h before and, simultaneously, fully completed the self-completion questionnaire and performed the anthropometric assessment. The project was approved by the Ethical Commission of the University of Porto, with the number CEFADE 25.2014. The principles of the Helsinki Declaration were respected and the workers under analysis accepted participation in this study through informed consent, after having the purpose and methods involved in the study explained to them individually. [18].

Data were collected by a nutritionist, and those relating to food intake in the form of a survey of the previous 24 h were entered into the eAT24 software (https://ispup.up.pt/en/dietary-record-coding/, accessed on 23 January 2024) of the IAN-AF [20]. Food quantification was carried out based on reported quantities or household measures or, if this information was not available, the default quantity defined by the program was considered. To help with using the software and when deciding on the omission of food items and/or reported amounts, a Procedures Manual was prepared, suitable for this work. Through the eAT24 software, it was possible to obtain information on the food consumption of the participants in terms of macro- and micronutrients. That information was compiled in a Microsoft Excel file together with the remaining information collected and selected within the scope of this study, to carry out the predicted calculations and, subsequently, the statistical treatment.

### 2.1. Adherence to MD

In order to evaluate MD adherence, we initially created an index that considered eight components (monounsaturated and saturated fat ratio, alcoholic beverage, pulse, cereal, fruit, vegetable, meat, and milk and dairy consumption) [21]. Currently, there are about fourteen different indices that consider different amounts of different food and/or nutritional components and that were created for application in different population groups [22,23]. One of them is the Mediterranean Diet Score (MDS) [15], which considers nine food items assigning a score of zero or one point for each of them. For the food groups associated with health benefits (vegetables, pulses, fruits, nuts, cereals, and fish), a score of one is assigned to participants whose consumption is equal to or greater than the median and, on the contrary, a score of zero to participants whose consumption is below the median. The food groups associated with less benefit to health (meat and derivatives, milk, and dairy products) are assigned a score of one when consumption is below the median and equal to or above the median, the score is zero. For ethanol, a score of one is considered when intake is between 10 and 50 g/day or between 5 and 25 g/day for men and women, respectively. The sum of the scores assigned to each food group ranges from zero (minimum adherence) to nine (maximum adherence) [7,14]. This score is considered valid to assess MD adherence [23]. The use of these indices becomes relevant to assess adherence to this dietary pattern and better understand the population’s dietary trends, relating them, for example, to other health factors.

The level of adherence to the MD of each participant was calculated based on the surveys in the previous 24 h. Each participant’s survey was inserted into the eAT24 software, from which the individual food consumption of each food group and subgroup was obtained, also obtaining the same information in the form of macronutrients. To calculate the MD adherence of each participant, only the consumption of the subgroups of vegetables, fruits and dried fruits, pulses, cereals, fish, meat, poultry, and dairy products was considered, as well as the consumption of macronutrients for the fat and ethanol groups, as described in Table 1. Each food group was given a score of zero or one, depending on how it corresponded to the median recommended intake adjusted for gender. For the food groups associated with health benefits (vegetables, pulses, fruits, cereals, fish, and the ratio of monounsaturated to saturated fats), consumption below the median was assigned a classification of zero and one if consumption was equal or superior to the median. For food groups whose consumption is associated with negative health implications (meat and dairy products), consumption equal to or less than the median was assigned a score of one, and consumption higher was assigned a score of zero. For ethanol, a value of one was attributed to men who consumed between 10 and 50 g per day and to women who consumed between 5 and 25 g per day. MD adherence resulted in the sum of all previous scores and ranged from zero (minimum adherence) to nine (maximum adherence) [15,16].

### 2.2. Food Intake Carbon Footprint Calculation

There is a variety of tools to measure environmental impact, and these can be classified into procedural tools or analytical tools. Analytical tools focus on the technical aspects of the analysis and include, among others, the life cycle assessment (LCA), which measures the environmental impact through the resources used during the life of a product, from the raw material that it originates from to its production and elimination [24]. LCA is a systematic, phased approach consisting of five components: goal and objective, setting, inventory analysis, impact assessment, and interpretation [25]. The use of LCA to document and monitor environmental performance at the corporal level is often limited to a few selected impact categories, such as carbon footprint and blue water footprint [26]. According to ISO/TS 14067:2013, the carbon footprint (CF) of a product is defined as the “sum of greenhouse gas emissions (GGE) and removals in a product system, expressed as carbon dioxide equivalents (CO_2_eq) and based on a LCA using the single impact category of climate change” and CO_2_eq as a “unit for comparing the radiative forcing of a greenhouse gas to that of carbon dioxide” [27]. Despite the fact that several indicators have emerged to measure GGE, CF is the most popular and widely used to raise awareness about environmental impact [28].

Food production is one of the main contributors to GHG emissions in the world and it is known that this parameter is closely related to the dietary pattern adopted by the population since the impact generated on the environment varies depending on the food produced [29].

The carbon footprint of food intake was calculated from the information collected in the surveys of 24 h recalls and this tool was considered valid for this analysis [30]. To calculate the CF of each participant’s food intake, a website [31] was used, which uses a database of GHG emissions per food, with average data, generally at a global level, and some specific local values from the United Kingdom. The data obtained did not consider emissions at the level of retail and food preparation due to the difficulty in calculating these values and their attribution to individual foods. The values presented were displayed in CO_2_eq per kilogram of food (CO_2_eqKg^−1^), which corresponds to the total impact of the product based on all greenhouse gas emissions released, since, although CO_2_ is the main greenhouse gas, there are others such as methane and nitrous oxide that must be considered to properly assess the global warming potential of a product. For each food group obtained from the surveys of 24 h recalls, an average CO_2_eqKg^−1^ of each food item available on the website [31] was determined. The average CO_2_eqKg^−1^ value of each food group was multiplied by the amount of food consumed by each person. The individual daily CF is the sum of the CO_2_eqKg^−1^ of each food group consumed. Because it is a commonly used indicator to measure the environmental impact, with particular use in the environmental impact on the food chain [32], the use of CF was considered sufficient for the purpose of this work. To achieve the objectives of this work, the analysis of the carbon footprint based on GHG emissions was considered sufficient, since an analysis of other variables, such as the water and ecological footprint, could become complex and difficult to carry out [28].

### 2.3. Other Variables and Their Relationships with MD Adherence and CF of Food Consumption

From the MD adherence and food consumption CF calculated for each participant, it was intended to understand if there was any relationship between both variables. From the collected self-administered questionnaire, sociodemographic questions and questions related to eating habits and lifestyles were selected. All sociodemographic questions in the self-administered questionnaire were considered in this study. From the anthropometric measurements taken, the Body Mass Index (BMI) was calculated, which was the only health variable used to relate the MD adherence and food consumption CF of the participants. From the set of questions related to eating habits and lifestyles, some questions were selected to understand what relationship exists between these and adherence to the MD adherence and food consumption CF of each participant, such as the type of activity they perform during work; meals at the workplace and how often; use of the bar; vending machine use and consumption of food brought from home; where they usually have lunch; frequency of consumption of fresh fruit, soup, and vegetables at work; amount of water consumed per day; frequency of consumption of alcoholic beverages, their type and quantity; self-perception of adopting a healthy diet; distance between home and work; time and mode of commuting; the practice of programmed physical activity outside the workplace; the description of the state of health; the prevalence of chronic diseases; and performance at work in relation to energy levels, mood, concentration, stress levels and productivity.

The tools used to measure MD adherence and food consumption have limitations that depend in part on the constructs and scoring of diet quality and the relationship between them [32].

### 2.4. Statistical Methods

Microsoft Office Excel version 2401 was used to calculate MD adherence and food consumption CF, and all data were analyzed in IBM SPSS Statistics, version 27. Descriptive statistics consisted of relative (%) and absolute (n) frequencies for categorical variables and mean ± standard deviation, median, maximum, and minimum for quantitative variables. The normality of cardinal variables was analyzed using symmetry and flattening coefficients. *T*-tests for independent samples, ANOVA, and Spearman and Pearson correlation coefficients were used to relate the variables. Significance was set at the 0.05 level.

### 2.5. Ethical Issues

This project was approved by the Ethical Commission of the University of Porto, with the number CEFADE 25.2014. The principles of the Helsinki Declaration were respected and the workers under analysis accepted participation in the study through informed consent, after having the purpose and methods involved in the study explained to them individually.

## 3. Results

From 368 participants eligible for this study, it was found that the average age was 43.3 ± 10.8 years, aged between 21 and 80 years, the majority were female (66.0%), and about 64.7% of the sample was married. Of the total, 77.7% had higher education, of which 11.4% was in health sciences, 9.2% in humanities, 10.3% in natural sciences, 9.5% in physical sciences, 23.6% in social sciences, and 13.6% in technological sciences. It was found that the average number of years working at the University was 16.0 ± 11.1 years, 67.9% of the sample had a non-teaching job, and 1.6% had teaching and non-teaching jobs at the Institution (Table 2).

### 3.1. MD Adherence and Food Intake CF

The study sample had a mean MD score of 3.88 ± 1.49 on a scale between 0 and 9. It was found that only 13.2% of the study sample showed an MD score greater than 6, on a scale of 0 to 9.

Food consumption CF calculated for the sample had an average of 8146 ± 3081 CO_2_eqKg^−1^.

There was a weak significant correlation between the MDS score and food consumption CF (r = 0.142, *p* = 0.006), which means that higher MDS scores are associated with higher food consumption CF.

### 3.2. Relationship between Sociodemographic Characteristics, Lifestyle, and Eating Habits with MD Adherence

It was found that there were statistically significant correlations between the participants who considered bringing healthy options from home (*p* = 0.043), the practice of programmed physical activity (*p* = 0.019), and MD adherence. There was also a weak positive correlation between age (r = 0.149; *p* = 0.004), consumption of alcoholic beverages (r = 0.132; *p* = 0.011), frequency of consumption of fresh fruit (*p* = 0.003), soup (*p* < 0.001), and vegetables at work (*p* = 0.042), and MD adherence. On the other hand, there was a weak negative correlation between self-perceived performance at work in relation to productivity (r = −0.103; *p* = 0.047) and MD adherence. It was found that there were no statistically significant differences between MD adherence and area of field and performance at work in terms of energy levels, humor/good mood, concentration, stress levels, or productivity (Table 3).

### 3.3. Relationship between Sociodemographic Characteristics, Lifestyle, and Eating Habits with Food Intake CF

Regarding the relationship with CF of food intake, it was found that there were statistically significant differences between people who eat lunch at home (*p* = 0.008), those who consider eating healthy (*p* = 0.002), the means of travel to work (*p* = 0.038), and scheduled physical activity (*p* = 0.047). A positive correlation was also observed between the CF of food intake and the time working in the University (r = 0.145; *p* = 0.005), the frequency of consumption of fresh fruit (r = 0.316; *p* < 0.001), soup (r = 0.144; *p* = 0.006), and vegetables at work (r = 0.179; *p* < 0.001), and description of health status (r = 0.144; *p* = 0.010). On the other hand, a negative correlation was found between the food consumption CF and the self-perception of stress levels in performance at work (r = −0.114; *p* = 0.028). It was found that there were no statistically significant differences between CF of food consumption and area of field and performance at work in terms of energy levels, humor/good mood, concentration, stress levels, or productivity (Table 4).

## 4. Discussion

The results obtained in this study reflect, on one hand, the low MD adherence (average of 3.88 of the MDS) of the sample, as well as the high CF of food consumption (8146 ± 3081 CO_2_eqKg^−1^). A recent study that used data from the IAN-AF concluded that 19.6% of Portuguese adults had an MDS equal to or greater than 6 [33], a better result when compared with the data presented in this study (13.2% of the sample had an MDS equal to or greater than 6). On the other hand, the data obtained describe a CF of food consumption twice as high as that reported in Brazil (4489 g CO_2_eq) [34], France (4170 g CO_2_eq) [35], and the United States of America (4720 g CO_2_eq) [36]. These results are different from the conclusion of García, S. et al., which showed that higher MD adherence is related to lower odds of dietary carbon dioxide emissions [37]. This discrepancy can be explained by the fact that global data were used, and some specific values from the United Kingdom. Additionally, a positive and weak correlation was observed between the previous variables (r = 0.142; *p* = 0.006), suggesting that greater MD adherence is associated with greater food intake CF. These results may be related either to the existence of products of animal origin in this dietary pattern [38], or to the actual consumption habits of the sample under study. Although there are few studies that have studied the relationship between these two variables, a study carried out with children in Spain also concluded that there is a weak positive correlation between MD adherence and food intake CF, arguing the weight that animal sources can have on the environmental impact, even though the MD’s recommendation for meat consumption is moderate [39]. Other studies have concluded that dietary patterns without food of animal origin have a smaller negative impact compared to the MD [39], but diets richer in food sources of animal origin seem to have a greater environmental impact compared to the MD [38]. It has been demonstrated that meat and animal products have a greater CF compared with fruits and vegetables [40], but further research should be conducted to analyze the contribution of animal products to the CF of the diet and the confounding effect of each food group on the verified association, in this sample and in other populations [41].

It was also observed that participants with programmed physical activity had, on average, higher MDSs (*p* = 0.019), thus showing greater MD adherence, results consistent with those observed in the IAN-AF [42]. These results are particularly important since it was previously shown that higher levels of physical activity are related to greater adherence to the MD and that, in turn, both these variables represent higher levels of health-related quality of life [43].

Older participants had, on average, greater adherence to the MD as it was found that there was a statistically significant positive correlation between the MDS and the participants’ age (r = 0.149; *p* = 0.004). This result is consistent with others reported in Portugal [42,44] and reflects the influence of socioeconomic, cultural, and lifestyle factors on the eating habits of younger people [45,46].

The MDS includes a score for ethanol consumption [14] that is consistent with the positive, statistically significant correlation observed between the MDS and the consumption of alcoholic beverages (r = 0.132; *p* = 0.011).

Consumption of fresh fruit (r = 0.152; *p* = 0.003), soup (r = 0.106; *p* < 0.001), and vegetables (r = 0.132; *p* = 0.011) in the workplace demonstrated a positive, statistically significant correlation with the MDS, indicating that the sample complies with these MD principles [25]. Thus, it should be emphasized that the food offered tends to promote the MD principles in food services, and along with other strategies to promote this dietary pattern, it must be adopted to increase adherence, as suggested by the literature [47].

With regard to the relationship between the food intake CF and the remaining variables under study, it was found that those who have lunch at home had, on average, a higher food consumption CF than those who do not (*p* = 0.008), which corroborates the conclusion of a recent study that demonstrated that frequently cooking dinner at home is associated with higher GHG emissions associated with food consumption [48]. A systematic review observed that meals produced at home may have higher GHG emissions compared to those made in food services/restaurants, highlighting that strategies such as reducing the consumption of ingredients of animal origin, raising consumer awareness of the environmental impact of food consumption, and formulating more sustainable menus should be adopted to improve this indicator [49]. The acquisition of locally produced raw materials can also be another strategy to minimize the negative impacts that the food sector may have [50].

Other measures can be adopted to minimize non-food consumption-related CF in the context of food service units, such as scheduling specific hours for the operation of high-energy-cost equipment, keeping equipment in good working conditions and hygiene, the adoption of measures and practices that avoid the presence/infestation of pests, and the training of all personnel involved [51]. Together, these strategies can facilitate competitiveness with food brought from home from the point of view of environmental impact, enable the adoption of healthier eating habits by the public, and enable cost reduction in the context of food service units.

Taken together, these results can be particularly interesting from the perspective of encouraging the use of food services, in this case, including the offer available at this University setting campus, as places that can facilitate access to healthy food offers and help to promote and raise awareness towards more sustainable eating habits.

Although we used a validated tool [23], it was not possible to use one that could be validated specifically in the Portuguese population. Because of that, other studies could be carried out in order to provide a valid tool to calculate MD adherence in the Portuguese population. We recognized other limitations such as the use of reference data to calculate GHG emissions from food consumption from the United Kingdom, which may not represent the reality of this indicator in Portugal, hampered by the scarce information available at the national level. Other data on food consumption, such as the food waste generated, the origin of the food, and the cooking methods, were not collected and, therefore, their analysis was not considered in this work. The use of other indicators such as the water footprint or the ecological footprint could have added useful and relevant information in this context. There are few works that relate food consumption CF and MD adherence and, therefore, other studies should be conducted to study the relationship between these two variables and understand the existing relationships between them, to compare with the results obtained here. Data regarding sustainable indicators of food consumed in Portugal must be collected and made available in order to better compare its environmental impact. Additionally, the relationship between different food patterns and the CF should be addressed to improve the guidelines for healthy and sustainable diets in this dimension.

Nevertheless, this was one of the few works in Portugal to link MD adherence to food consumption CF, which allowed reflection on these two dimensions, allowing us to bring in strategies that reconcile the dimension of sustainability combined with the promotion of healthy eating habits from the perspective of food services. More work is needed to measure and compare data on the environmental impact of food outlets and the food consumption of their users that can serve as a basis for promoting policies and initiatives that simultaneously encourage healthy eating habits in their users and raise awareness of environmentally sustainable lifestyles.

## 5. Conclusions

Most of the participants had a low MD adherence, 13.2% of the sample had an MDS equal to or greater than six, on a scale from zero to nine, as well as an average food intake CF of 8146 ± 3081 CO_2_eqKg^−1^, considered a high value when compared to other countries. This study concluded that high MD adherence corresponds to a greater food intake CF, although a weak, statistically significant positive correlation was found (r = 0.142, *p* = 0.006). These results suggest that reducing the consumption of animal-based products should be encouraged and highlight that having lunch at home is associated with higher food intake CF than those who do not have lunch at home, which represents an opportunity for the University food services to both encourage healthy food options and help lower the CF of food intake of the people working there. Other measures like encouraging fruit, vegetable, soup, and water consumption in the University setting can help increase MD adherence in this setting. Further research should be conducted to analyze the contribution of animal products to the CF of the diet and the confounding effect of each food group on the verified association, in this sample and in other populations, and to provide data regarding sustainable indicators of food consumed in Portugal in order to better compare its environmental impact.

## Figures and Tables

**Table 1 nutrients-16-00635-t001:** Description of food and nutrient groups considered for calculating MD adherence.

Food Groups According to eAT24 Software	Food Groups Considered to Measure MD Adherence
Vegetables, soup	Vegetables
Nuts and seeds. Fresh fruit. Processed fruit.	Fruits and Nuts
Pulses	Pulses
Pasta. Rice. Bread and toast. Flour. Children’s cereals. Breakfast cereals.	Cereals
Fish. Crustaceans and mollusks. Processed fish.	Fish
Red meat. Guts. Charcuterie.	Meat
White meat.	Poultry
Milk. Human milk. Infant formulas. Cream and dairy. Yogurts. Cheese.	Dairy
Ratio between consumption of monounsaturated fatty acids and consumption of saturated fatty acids	Fats
Ethanol	Ethanol

**Table 2 nutrients-16-00635-t002:** Description of sociodemographic and health characteristics of the participants.

	Number of Participants	% of Participants (%)
Sex		
- Female	243	66.0%
- Male	125	34.0%
Weight (Kg)		
- [0–47]	167	45.4%
- [47–87]	157	42.7%
- [87–120]	44	12.0%
IMC		
- Underweight (<18.5 kg/m^2^)	9	2.5
- Normal (18.5–24.9 kg/m^2^)	191	52.6
- Overweight (>24.9 kg/m^2^)	163	44.9
Marital status		
- Single	102	27.7%
- Married/De facto union	238	64.7%
- Separated/Divorced	27	7.3%
- Widowed	1	0.3%
Academic qualifications		
- 1st Cycle	3	0.8%
- 2nd Cycle	5	1.4%
- 3rd Cycle	9	2.4%
- High School	65	17.7%
- Bachelor’s Degree	111	30.2%
- Master’s Degree	60	16.3%
- Doctoral Degree	92	25.0%
- Post-Doctoral Studies	23	6.3%
Study area (n = 285)		
- Health sciences	41	11.1%
- Humanities	34	9.2%
- Natural sciences	38	10.3%
- Physical sciences	35	9.5%
- Social sciences	87	23.6%
- Technological sciences	50	13.6%
Professional Class:		
- Teacher	112	30.4%
- Non-teacher	250	67.9%
- Both	6	1.6%
Total	368	

**Table 3 nutrients-16-00635-t003:** Relationship between MD adherence and sociodemographic characteristics, lifestyle, and eating habits of the participants.

		MD Adherence	
		Mean	*p*
Gender	Female	3.82	0.315
	Male	3.98	
Scientific Area of work	Health Sciences	3.71	0.674
	Social Sciences	3.76	
	Humanities	3.85	
	Natural Sciences	4.00	
	Physical Sciences	3.97	
	Technological Sciences	4.14	
	No	3.88	
Do you use the snack-bar?	Yes	3.92	0.301
	No	3.74	
Do you consider that the snack-bar has healthy options?	Yes	4.19	0.312
	No	3.77	
Do you use vending machines?	Yes	3.83	0.397
	No	3.96	
Do you consider vending machines has healthy options?	Yes	3.83	0.833
	No	3.91	
Do you bring food from home?	Yes	3.88	0.976
	No	3.87	
Do you consider that the food you bring from home is healthy?	Yes	3.00	0.043
	No	3.83	
Do you have lunch at the University food services?	Yes	4.04	0.131
	No	3.79	
Do you have lunch that the snack-bars/restaurants near the University?	Yes	4.08	0.187
	No	3.82	
Do you usually bring lunch from home?	Yes	3.74	0.077
	No	4.02	
Do you usually have lunch at home?	Yes	3.64	0.357
	No	3.91	
Do you usually drink beer?	Yes	4.13	0.113
	No	3.81	
Do you usually drink wine?	Yes	4.09	0.052
	No	3.77	
Do you usually drink distilled beverages?	Yes	4.45	0.191
	No	3.86	
Do you consider you eat healthy?	Yes	3.93	0.167
	No	3.64	
How do you go to work?	Walking	3.90	0.977
	Personal Transport	3.87	
	Public Transport	3.89	
Do you take any programed physical activity?	Yes	4.06	0.019
	No	3.70	
Do you have any chronic illness?	Yes	4.01	0.292
	No	3.83	
How old are you?		0.149	0.004
How long do you work at the University?		0.032	0.542
What is the amount of water you drink during the day?		0.006	0.908
How long does it get to get from home to work?		0.027	0.610
BMI		0.032	0.542
At your workplace, how often do you usually eat fresh fruit?		0.152	0.003
At your workplace, how often do you usually eat soup?		0.264	<0.001
At your workplace, how often do you usually eat vegetables, except in the soup?		0.106	0.042
How often do you drink alcoholic beverages?		0.132	0.011
How far do you live from your workplace?		0.030	0.566
How do you describe your health, generally?		−0.017	0.750
How would you describe your performance at work in terms of energy levels?		−0.058	0.268
How would you describe your performance at work in terms of humor/good mood?		−0.062	0.232
How would you describe your performance at work in terms of concentration?		−0.073	0.162
How would you describe your performance at work in terms of stress levels?		−0.021	0.693
How would you describe your performance at work in terms of productivity?		−0.103	0.047

**Table 4 nutrients-16-00635-t004:** Relationship between food intake CF and sociodemographic characteristics, lifestyle, and eating habits of the participants.

		Food Intake CF (CO_2_eqKg^−1^)	
		Mean	*p*
Gender	Female	8571	0.089
	Male	7955	
Area of field	Health Sciences	7790	0.456
	Social Sciences	7930	
	Humanities	8005	
	Natural Sciences	8279	
	Physical Sciences	8792	
	Technological Sciences	8878	
	No	8182	
Do you use the snack-bar?	Yes	8087	0.439
	No	8372	
Do you consider that the snack-bar has healthy options?	Yes	6892	0.084
	No	8141	
Do you use the automatic food machine?	Yes	7940	0.059
	No	8571	
Do you consider that the automatic food machine has healthy options?	Yes	7843	0.477
	No	8356	
Do you bring food from home?	Yes	7625	0.226
	No	8233	
Do you consider that the food you bring from home is healthy?	Yes	5446	0.448
	No	6995	
Do you have lunch at the University food services?	Yes	8058	0.648
	No	8214	
Do you have lunch at the snack-bars/restaurants near the University?	Yes	7887	0.402
	No	8229	
Do you usually bring lunch from home?	Yes	8228	0.693
	No	8100	
Do you usually have lunch at home?	Yes	9270	0.008
	No	8000	
Do you usually drink beer?	Yes	8014	0.703
	No	8197	
Do you usually drink wine?	Yes	8564	0.081
	No	7958	
Do you usually drink distilled beverages?	Yes	9070	0.323
	No	8134	
Do you consider you eat healthy?	Yes	8402	0.002
	No	7176	
How do you go to work?	Walking	8562	0.038
	Personal Transport	7960	
	Public Transport	8381	
Do you take any programed physical activity?	Yes	8491	0.047
	No	7847	
Do you have any chronic illness?	Yes	8284	0.645
	No	8117	
How old are you?		0.043	0.407
How long do you work at the University?		0.145	0.005
What is the amount of water you drink during the day?		0.086	0.101
How long does it get to get from home to work?		0.033	0.523
BMI		−0.070	0.178
At your workplace, how often do you usually eat fresh fruit?		0.316	<0.001
At your workplace, how often do you usually eat soup?		0.144	0.006
At your workplace, how often do you usually eat vegetables, except in the soup?		0.179	<0.001
How often do you drink alcoholic beverages??		0.032	0.535
How far do you live from your workplace?		−0.015	0.783
How do you describe your health, generally?		0.134	0.010
How would you describe your performance at work in terms of energy levels?		0.096	0.067
How would you describe your performance at work in terms of humor/good mood?		0.045	0.391
How would you describe your performance at work in terms of concentration?		−0.004	0.941
How would you describe your performance at work in terms of stress levels?		−0.114	0.028
How would you describe your performance at work in terms of productivity?		−0.026	0.614

## Data Availability

Data is contained within the article.

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
