# Peer review of "Adherence to the Mediterranean Food Pattern and Carbon Footprint of Food Intake by Employees of a University Setting in Portugal"

_nutrients, 2024, doi:10.3390/nu16050635_

Round 1
Reviewer 1 Report (New Reviewer)
Comments and Suggestions for Authors
1. Is there any association between food production and Mediterranean food pattern? It should be more concise in the Introduction.
2. What is the carbon footprint of food intake? It should be contained in the Introduction.
3. The introduction about MD is too lengthy. It needs rewritten and should be limited in reasonable words.
4. The objective aim of this study should be introduced in detail.
5. The characteristics about participants should be placed in the results.
6. The ethical declaration should be set as a single part.
7. How did the related factors influence the associations between MD and carbon footprint of food intake, such as having lunch at home, healthy eating and approaches to work? Analysis of covariates is needed.
Comments on the Quality of English Language
Moderate editing of English language required.
Author Response
- Is there any association between food production and Mediterranean food pattern? It should be more concise in the Introduction.
The association between MFP and Food production is evident. We add this information in the introduction, more clearly.
The Mediterranean Diet (MD) presents a lower ecological footprint comparing to Western dietary pattern [19,20,21], due to the predominance of plant-based foods, local and seasonal crops, and a sustainable food culture, that embraces important physical, sociocultural, economic, and environmental benefits [14]; the reason why the adoption of MD triggers local food production.
- What is the carbon footprint of food intake? It should be contained in the Introduction.
In the introduction we have this data:
According to ISO/TS 14067:2013 the carbon footprint (CF) of a product is defined as the “sum of greenhouse gas emissions (GGE) and removals in a product system, expressed as carbon dioxide equivalents (CO2eq) and based on a LCA using the single impact category of climate change” and CO2eq as an “unit for comparing the radiative forcing of a greenhouse gas to that of carbon dioxide” [32].
In methods we have this information:
To calculate the CF of each participant's food intake, a website [40] was used, which uses a database of GHG emissions per food, with average data, generally at a global level, and some specific local values, from United Kingdom.
One of the aims of the study was to calculate the carbon footprint of food intake of our sample.
- The introduction about MD is too lengthy. It needs rewritten and should be limited in reasonable words.
Introduction was rewritten.
- The objective aim of this study should be introduced in detail.
More detail was added in the aim of the study:
In a representative sample of university employees, the aims of this study were:
- to evaluate mediterranean diet adherence of food consumption;
- to analyze the Carbon Footprint of food consumption;
- to understand the relationship between MD adherence CF of their food intake in this population.
- The characteristics about participants should be placed in the results.
This information is already placed in the results (lines 298-303).
- The ethical declaration should be set as a single part.
Information was added in a new sub-heading (2.5.):
The project was approved by Ethical Commission of the University of Porto, with the number CEFADE 25.2014. The principles of the Helsinki Declaration were respected and the workers under analysis accepted participation in the study through informed consent, after having the purpose and methods involved in the study explained to them individually.
- How did the related factors influence the associations between MD and carbon footprint of food intake, such as having lunch at home, healthy eating and approaches to work? Analysis of covariates is needed.
The associated factors that influence MD and CF were analysed on table 2 and 3, respectively. To MD we didn’t find differences between these factors. Differences were observed related to CF, for instance, people who consider having healthy eating or having lunch at home presented a higher CF of their intake.
However, according to our methods, we can’t explain how this happens. We verify that when one condition occurred the other too. Additional studies would be necessary to explain this association that we identify in this paper.
Moderate editing of English language required.
English language was revised.
Reviewer 2 Report (New Reviewer)
Comments and Suggestions for Authors
The objective of this study described in the abstract is to understand the relationship between the degree of adherence to the Mediterranean diet and the relationship 19 with the carbon footprint of their food consumption. In the introduction it is this AND to understand the degree of adherence to the MD diet. This should be synchronised. I also noticed spelling and grammatical errors in the introduction, so the paper needs to thoroughly and carefully reread to look of spelling and/or grammatical mistakes and not just left to the spellchecker. For example, Page 2 line 56, carious, which I presume that you meant to be various.
The introduction hasn’t established a motivation for the study. It might make more sense to compare the CFs of two different types of diets, or those that adhere to MD and those that don’t, but in its present form this is just a descriptive study.
The sample is a sample of academics, so what is the inferential population to which the results are intended to apply?
It might be easier for the reader if a scatterplot is provided to illustrate the correlation between MD and CF. The information in Table 2 was partitioned by gender. This may be interesting, but it is the first time that the reader is seeing this and not related to the aim of the study.
What statistical test was used obtain the p-value in table 3?
What is the implication of the main result of adherence to MD of 13.2% and CF of 8146 and a non-significant correlation between the two of 14.2%? On my reading there is no evidence to suggest that this study is necessary or that the result change anything. The paper’s main result is effectively a negative result, which has been published before so it is difficult to see how this manuscript adds anything to the body of knowledge.
Author Response
The objective of this study described in the abstract is to understand the relationship between the degree of adherence to the Mediterranean diet and the relationship 19 with the carbon footprint of their food consumption. In the introduction it is this AND to understand the degree of adherence to the MD diet. This should be synchronised. I also noticed spelling and grammatical errors in the introduction, so the paper needs to thoroughly and carefully reread to look of spelling and/or grammatical mistakes and not just left to the spellchecker. For example, Page 2 line 56, carious, which I presume that you meant to be various.
Introduction and objective were rewritten.
The paper was reread and spelling and grammatical mistakes were corrected.
The introduction hasn’t established a motivation for the study. It might make more sense to compare the CFs of two different types of diets, or those that adhere to MD and those that don’t, but in its present form this is just a descriptive study.
The study was, actually, an observational study and not a case-control or a randomized control trial. So, we analyse associations between variables and correlations, namely between MD score and carbon footprint.
In point 3.1. of results: MD Adherence and Food Intake CF we stated this association:
The study sample had a mean MD score of 3.88 ± 1.49 in a scale between zero and nine. It was found that only 13.2% of the study sample showed an MD score greater than 6, on a scale of 0 to 9. Food consumption CF calculated for the sample had an average of 8146 ± 3081 CO2eqKg-1.
There was a weak significant correlation between the MDS score and food consumption CF (r=0.142, p=0.006), which means that higher MDS score are associated with higher food consumption CF.
The sample is a sample of academics, so what is the inferential population to which the results are intended to apply?
This study were conducted in a representative sample of the population of academics and non-academic staff of one of the biggest public universities in Portugal. We haven’t the intention of generalizing the results to the general population. However, it is a group of workers and highlights for other studies in other populations from this could be important.
It might be easier for the reader if a scatterplot is provided to illustrate the correlation between MD and CF. The information in Table 2 was partitioned by gender. This may be interesting, but it is the first time that the reader is seeing this and not related to the aim of the study.
In spite of gender issue is not included in the objective, it was referred to a better characterization of the sample.
We didn’t included the scatterplot attending to the strength of correlation: (r=0.142, p=0.006).
What statistical test was used obtain the p-value in table 3?
T-tests for independent samples, ANOVA and Spearman and Pearson correlation coefficients depending of type of variables (cardinal and ordinal; cardinal and nominal…).
What is the implication of the main result of adherence to MD of 13.2% and CF of 8146 and a non-significant correlation between the two of 14.2%? On my reading there is no evidence to suggest that this study is necessary or that the result change anything. The paper’s main result is effectively a negative result, which has been published before so it is difficult to see how this manuscript adds anything to the body of knowledge.
This paper highlights a weak positive correlation between adherence to the Mediterranean diet and carbon footprint, with statistical significance.
We commonly associate the Mediterranean diet with more sustainable food consumption, however, these results point to an opposite direction.
In the authors' opinion, a broader reflection and more in-depth studies are necessary that relate these two variables, as dietary recommendations that promote adherence to Mediterranean diet as healthy and sustainable diet (namely through dietary guides), do not seem to have as much evidence as expected.
Round 2
Reviewer 1 Report (New Reviewer)
Comments and Suggestions for Authors
1. The introduction about MD shoule be more concise. It is a introduction but not a review. Irrelevant information should be removed.
2. The characteristics about participants should be added in the results as a table, such as number, sex and body weight.
3. When some factors were identified to influence the associations, more analysis were needed to explian the phenomena as far as possible based on available results. Therefore, analysis of covariates is necessarily needed, and it should be discussed in the Dissusion.
Comments on the Quality of English Language
Minor editing of English language required
Author Response
Dear review
Thank you very much for your suggestions and comments. They were very useful to improve the quality of our paper.
- The introduction about MD shoule be more concise. It is a introduction but not a review. Irrelevant information should be removed.
The Introduction was shortened as suggested.
2. The characteristics about participants should be added in the results as a table, such as number, sex and body weight.
A table was included with socio-demographic and BMI / weight information.
3. When some factors were identified to influence the associations, more analysis were needed to explian the phenomena as far as possible based on available results. Therefore, analysis of covariates is necessarily needed, and it should be discussed in the Dissusion.
We believe associations between variables are all presented in Tables 3 and 4. We performed the additional statistical analysis, but the output didn't give additional insights, the reason why didn't include this information.
Despite the frequent association between MD and sustainability, our results point in the opposite direction. Additionally, from our knowledge other colleagues are obtaining similar results in other samples/populations (non-published - data in the review process for publication).
Reviewer 2 Report (New Reviewer)
Comments and Suggestions for Authors
The authors have made a good faith attempt to address my comments.
Author Response
Thank you very much for your time and comments that contributed to the improvement of the paper.
This manuscript is a resubmission of an earlier submission. The following is a list of the peer review reports and author responses from that submission.
Round 1
Reviewer 1 Report
Comments and Suggestions for Authors
Dear authors,
the paper presented to Nutrients has with the main focus on food patterns according to the relation with the profiles of Portuguese consumers.
The paper has high originality, and a positive contribution to the scholarship.
It has very high level of provided evidence and an appropriate sample.
The article has used a vast source of literature, showed empirical evidence in the case of Portugal. and has shown their results in comparison to other authors.
After the changes made according to the reviewers opinion the paper is much better and is now read yfor publishing.
kind regards, the reviewer.
Author Response
Thank you for your positive feedback!
Reviewer 2 Report
Comments and Suggestions for Authors
The corrections made improved the quality of the paper.
I ask the authors to correct the following further:
CO2e = CO2eq (throughout the text)
CO2e/Kg = CO2eq/Kg (throughout the text)
CO2eKg-1 = CO2eqKg-1 (throughout the text)
Line 102, 112, 118, 131, 143, 148, 150, 154, 159, 181, 189, 235, 322, 325, 326, 340, 373, 375, 380: You must add a space between the word and the bibliographic reference.
Line 300: productivity. (Table 2). = productivity (Table 2).
Line 315: productivity. (Table 3). = productivity (Table 3).
Author Response
Thank you for your comments! We changed the paper as you suggested and highlighted the changes in gree
Reviewer 3 Report
Comments and Suggestions for Authors
First of all, corrections are needed related to the way the text was edited: starting from line 139, almost all bibliographic references in the text need space in front, starting with section 2.2 the carbon dioxide formula is not written correctly, then in table 2 and table 3 for the question "how often do you consume alcoholic beverages" the question mark is double, the bibliography is written as a joke, it does not respect the format, it does not respect the scientific level of the journal, and at the end of the article the approval of the study should have been specified.
Regarding the scientific quality of the article, from my point of view, it does not rise to the level of the journal. Starting from the presentation of the methodology, the data are unclear and the message ambiguous (for example: The population under study is the human resources of the UP which, according to the time of data collection, accounted for 3307.7 full-time workers, of which 1750.1 were professors /researchers, 1551.6 179 were non-teachers /non-researchers and six members of government bodies), then we do not have as additional material the questionnaire used as the analysis method, but it is not sufficiently well described in the methodology either. The statistical processing of the data seems simplistic to me, no valuable correlations are made between the data (adherence to the Mediterranean diet and BMI groups, BMI groups and age categories), after reading the material I am not convinced if adherence to the Mediterranean diet was correctly calculated. In its current form, I cannot recommend the publication of the article.
Author Response
Thank you for your comments! We corrected the space between references and the text, and the carbon dioxide formula highlighted in green, as other reviewers had already mentioned that improvement. As for the question mark
As for the bibliography we changed as Nutrients rules, something we did before, but probably that change wasn´t saved. That, as well as the extra question mark in both Table 2 and 3 were changed and highlighted in blue.
As for the methodology, the discrimination of staff in University of Porto is described as is said in the original report, but to clarify this issue, we changed the presentation of data and highlighted in blue.
The Mediterranean diet score was calculated exactly as A. Trichopoulou (2003) did and it was considered a validated methodology, that’s why we decided to go in that direction. Unfortunately, we don’t have any validated tools to calculate MD adherence in Portugal, so we will also add that suggestion in the discussion, highlighting it in blue.
Although we couldn’t reach the expected results, we believe that is important to share it with the scientific community in other to emphasize this topic, even in other countries, promote the availability of other data, as well as sustainable indicators of food consumed in Portugal, to better compare this results in the future.
Round 2
Reviewer 3 Report
Comments and Suggestions for Authors
I don't think it's a significant improvement, I leave it up to the academic editor to decide if the study is sufficiently interesting and valuable for the level and requirements of the journal. The writing of the bibliography, which is not exactly in line with the journal's requirements, was treated in a superficial way, what more can I say about the seriousness in improving the scientific value. I maintain my opinion related to the scientific quality of the manuscript.
Author Response
Dear reviewer
We corrected the references suggested by the academic editor and we are
sorry that you believe that the reference section was treated in a superficial way.
We would like to thank you for your time analyzing our work.